# Metapopulation Patterns of Iberian Butterflies Revealed by Fuzzy Logic

**DOI:** 10.3390/insects12050392

**Published:** 2021-04-28

**Authors:** Antonio Pulido-Pastor, Ana Luz Márquez, José Carlos Guerrero, Enrique García-Barros, Raimundo Real

**Affiliations:** 1Biogeography, Diversity and Conservation Research Team, Departamento de Biología Animal, Universidad de Málaga, 29071 Málaga, Spain; antonio.pulido@juntadeandalucia.es (A.P.-P.); rrgimenez@uma.es (R.R.); 2Laboratorio de Desarrollo Sustentable y Gestión Ambiental del Territorio, Instituto de Ecología y Ciencias Ambientales, Facultad de Ciencias, Universidad de la República, 11400 Montevideo, Uruguay; jguerrero@fcien.edu.uy; 3Departamento de Biología, Universidad Autónoma de Madrid, 28049 Madrid, Spain; garcia.barros@uam.es

**Keywords:** biogeography, dark biodiversity, ecological connectivity, favorability function, papilionoidea

## Abstract

**Simple Summary:**

We tested the usefulness of new analytical tools such as fuzzy set theory to reveal the hidden internal complexity of the geographic distribution of species. Fuzzy set theory replaces the crisp notion of presence/absence of a species, typical of species distribution atlases, with the fuzzy notion of favorability for the species occurrence. Species distribution ranges are then revealed as more complex than recorded presences suggest, and metapopulation theory, which predicts fragmented favorable patches with connectivity among them, can be operationally analyzed. We identified the favorable patches for 222 butterfly species in the Iberian Peninsula using high values of favorability. We calculated the cost of reaching any part of the territory from a favorable patch using low values of favorability and distance and computed the inverse as connectivity. Some of the favorable territories can be vacant patches but also belong to the metapopulation structure, as they may be recolonized. This information is relevant for territory management and biodiversity conservation, serving to justify the protection of new areas or the modification of contours of reserves based on the role they play for the populations of interest.

**Abstract:**

Metapopulation theory considers that the populations of many species are fragmented into patches connected by the migration of individuals through an interterritorial matrix. We applied fuzzy set theory and environmental favorability (F) functions to reveal the metapopulational structure of the 222 butterfly species in the Iberian Peninsula. We used the sets of contiguous grid cells with high favorability (F ≥ 0.8), to identify the favorable patches for each species. We superimposed the known occurrence data to reveal the occupied and empty favorable patches, as unoccupied patches are functional in a metapopulation dynamics analysis. We analyzed the connectivity between patches of each metapopulation by focusing on the territory of intermediate and low favorability for the species (F < 0.8). The friction that each cell opposes to the passage of individuals was computed as 1-F. We used the r.cost function of QGIS to calculate the cost of reaching each cell from a favorable patch. The inverse of the cost was computed as connectivity. Only 126 species can be considered to have a metapopulation structure. These metapopulation structures are part of the dark biodiversity of butterflies because their identification is not evident from the observation of the occurrence data but was revealed using favorability functions.

## 1. Introduction

The populations of many species can be fragmented by local extinction processes derived from certain stochastic natural phenomena: predation, resource depletion, disease, adverse weather, fires, or other disturbances [1]. Consequently, the continuity of species ranges is often broken, which results in a spatial pattern fragmented in more or less extensive patches [2]. In addition, human being presence pervades throughout the world, modifying and fragmenting the distribution ranges of many species. When a species is found in such fragmented plots and a certain level of connectivity exists among them, its distribution pattern constitutes a metapopulation, understanding it as a population of populations [3].

The concept of metapopulation has different definitions. It can be described as a population spatially organized into subpopulations more or less connected by migratory channels [1,4], as a spatially structured population that persists over time, as a set of local populations with limited dispersion among them [5], or as a population fragmented into a series of patches, embedded in a territorial matrix where the presence of the species is less favorable, that mutually maintain each other through the migration of individuals [6].

Understanding the ecology of metapopulations is important for biodiversity conservation science and has been extensively studied in recent decades [2,7,8,9]. Most authors were interested in the persistence of the metapopulation as a whole over time. They focused mainly on the size of the plots (local population centers) and the distance between them (connectivity/isolation), but also analyzed the role of the characteristics of patch and matrix habitats [2]. Patch size is considered mainly related to the processes of stochastic extinction. The characteristics of matrix habitats is related, together with the distance between patches, to connectivity, defined as the processes of emigration-immigration that allow the recolonization of currently unoccupied but suitable habitat patches [10,11], thus contributing to the persistence of the whole metapopulation. All this is known as Metapopulation Dynamics [12]. The key concepts are those of size of favorable patches and connectivity among them, and both concepts are fuzzy, naturally calling for the application of the fuzzy set theory [13] and the subsequent fuzzy logic to them. Fuzzy set theory and the use of some fuzzy concepts such as favorability [14] have previously been used in biogeographical studies [15,16].

Butterflies have undergone a remarkable change in species distributions over the last decades, including range fragmentation for many species [17,18,19]. Caterpillars have a role in the maintenance of fragmented patterns due to their limited mobility, while the adult butterflies have the ability to fly, being able to disperse. This makes them an indicator group of ecosystem quality, habitat transformation, and species responses to climate change and land use modification [20,21,22].

The Iberian Peninsula is one of the most species-rich areas for butterflies in Europe [23]. The distribution of Iberian butterflies is well known in general [24], although some areas have been insufficiently sampled [25]. Biogeographical studies of Iberian butterflies focused on geographical trends of species richness [21,26,27], distribution modeling for several species [22,28], and the analysis of source-sink dynamics [29]. However, the metapopulation structure of Iberian butterfly species remains poorly known.

The aim of this paper was to evaluate fuzzy set theory as a tool to reveal the favorable areas for species, either occupied or unoccupied, and thus to manifest hidden species metapopulation patterns, using Iberian butterflies as a case study. We also used fuzzy logic operators to identify the connectivity pattern between favorable patches, which contribute to migration processes and recolonization dynamics. We tested the applicability of fuzzy logic to assess metapopulation dynamics operationally over vast territories.

## 2. Materials and Methods

### 2.1. Study Area

The Iberian Peninsula is the westernmost of the Mediterranean peninsulas of Europe (Figure 1). It comprises about 580,000 km^2^ and two main countries, Spain and Portugal, in addition to Andorra and Gibraltar. This area was divided into 6040 grid cells of 10 × 10 km UTM (Universal Transverse Mercator). It is a highly mountainous territory of 600 m of mean elevation, with two plateaus and several mountain systems mainly going from west to east. These mountains exceed 2000 m in many parts and some summits reach over 3000 m. There is a clear lithological differentiation between the west (siliceous) and the east (limestone) of the peninsula [30] and remarkable climatic variations between the north (without summer drought) and the south (with hot and dry summers) [31]. There is also a noticeable difference between the annual precipitation volume between the west, facing the Atlantic Ocean, and the east, near the Mediterranean Sea. The western half has a more regular rainfall regime, with a minimum of about 450 mm per year and a maximum of about 2500 mm per year on average, although in the southwest quadrant (mountains of Cádiz, located in the south), up to 4500 mm has been recorded some years. The eastern Mediterranean part has a more irregular rainfall regime, with torrential episodes of a certain frequency, average volumes in the range 350–850 mm, with peaks of a much greater amount, and average annual precipitation ranges between 170–350 mm in the sub-desert area of the southeast.

### 2.2. Distribution Data and Explanatory Variables

The presence/absence of 222 species of butterflies present in the Iberian Peninsula were extracted from [32] and updated to 2009 by the authors. Iberian butterflies belong to the superfamily Papilionoidea, including the families Hesperiidae, Papilionidae, Pieridae, Nymphalidae, Lycaenidae. Species names from the original source have been kept for clarity. Some recent minor changes to species nomenclature [33,34] do not hamper the recognition of species identities.

We used the same variables previously used to assess the source-sink dynamics of butterflies in the Iberian Peninsula [29]. Although some authors only included climatic variables when modeling the distribution of species [35,36], it is unlikely that a species’ large-scale distribution will depend only on climate, as other environmental predictors such as topography, lithology, or human activity are likely also relevant and should be included in the biogeographical modeling of species distributions [37,38]. This is particularly true when analyzing the metapopulation structure of the species distribution, as the fragmented nature of species ranges often arises from the influence of human activities, which, in turn, may be affected by topography, spatial location, and other factors. This is why we used several types of explanatory variables, with 90 in total: 69 climatic, 6 topographic, 10 geological, four indicators of human activity, as well as two variables related to the spatial situation (see Appendix A). We selected these variables both because of their accessibility at our working scale and for their theoretical predictive power in relation to the group of species treated. We considered climate and topographic variables because they were shown to be the main drivers of the butterfly species richness [26]. Lithological variables are potentially relevant because of their bearing on the floral composition, given the herbivore and rather plant-specific nature of butterflies in their larval stage. Anthropogenic variables were considered important because humans have the capacity for modification of the natural environment [26,39]. In addition, purely spatial variables were used to describe the spatial structuring of the species, which allows the inference of the possible roles of population dynamics, dispersal capacities, and historical events on species distributions [40,41,42]. The spatial location also conditions climatic variables (Márquez, et al., 2004), so the true effect of climate must be assessed in the context of the spatial influences both on the species distribution and on climate. We used different sources for the climatic [31], topographic [43,44], lithological [30], and related with human activity [45,46] variables (See Appendix A).

### 2.3. Distribution Modelling

We used the same favorability models used to analyze the source-sink dynamics of butterfly species in the Iberian Peninsula [29]. We built an environmental favorability model for each butterfly species applying the following favorability function (FF) [14]:F = (P/(1 − P))/((n_1_/n_0_ + P/(1 − P)))(1)
where F is environmental favorability, P is the probability of occurrence, which we obtained using logistic regression, and n_1_ and n_0_ are the number of cells with reported presence and absence in the dataset, respectively. The FF reflects the degree (between 0 and 1) to which the local probability values differ from that expected according to the species’ prevalence, where F = 0.5 corresponds to a local probability value equal to the species prevalence in the Iberian Peninsula. The FF has been applied in biogeographical studies for different taxa and regions (for example, [47,48,49,50,51]).

Logistic regression is a supervised machine learning algorithm that related the species’ presence/absence on 10 × 10 cells of the Iberian Peninsula with the predictor variables (Appendix A). For each species, we performed a univariate logistic regression analysis of the effect of each variable separately on the species distribution and identified the significant explanatory variables. Then, we performed a selection of variables to build a parsimonious multivariate environmental probability model. We performed this selection following [52] (pp. 92–97) recommendation. As the number of variables was large and many of them were correlated, we controlled the increase in type I error due to the number of variables analyzed by evaluating the false discovery rate—FDR [53], accepting only the variables that were significant in a univariate logistic regression under an FDR of *q* < 0.05. Then, we performed a forward-backward stepwise multiple logistic regression on the remaining variables, which yielded increasingly complex and informative models, using the Akaike information criterion (AIC) to select the most parsimonious model [54]. The coefficients of this model were obtained by maximum likelihood estimation using an ascent gradient machine learning algorithm. The significance of these coefficients was tested using z values. When coefficients were non-significant, which was due to multicollinearity with the other variables in the model (e.g., [55], p. 442, [56,57]), we applied the ModelTrim function of the R package FuzzySim (R version 3.5.1) to remove variables until all coefficients for the variables were significant, which yielded the final model [58,59]. Collinearity only affects the transferability of the models [60]. As our models were used to describe the qualities of the observed data in the same territory where they were trained, collinearity was only a concern due to the redundancy of variables [60].

To evaluate the classification power of the models, we calculated their correct classification rate (CCR), sensitivity, specificity [61], under-prediction rate (UPR), over-prediction rate (OPR) [62], and their Cohen’s kappa [63], using the favorability value of F = 0.5 as classification threshold. To evaluate the discrimination capacity of the models, we evaluated the area under the curve (AUC) of the receiver operating characteristic [64]. The calibration of the models was evaluated using the Hosmer and Lemeshow calibration index (HL) using 10 bins of equal probability [52].

All analyses were performed in R 2.15.2 [65] with the packages FuzzySim [58,59] and ModEvA [58]. For more details on the modeling process, see [29].

Favorability is a fuzzy concept [14,66]. The favorability value for a species in a cell represents the degree of membership of the cell to the fuzzy set of favorable cells for the species, being the FF the membership function. This allows the incorporation of fuzzy logic operations to the spatial analysis of the species [67,68].

In contrast to the discrete nature of presences-absences mapping, the use of a continuous function sets gradual values for the entire territory under study. Each point of the territory had a value of the FF that was represented graphically in the entire study area and processed with the analysis tools available in geographic information systems (GIS) such as QGIS 2.18.4 [29].

### 2.4. Metapopulation Structure

After obtaining for each species the gradient of favorability (F) along the Iberian Peninsula, the sets of contiguous grid cells with high favorability (F ≥ 0.8) were considered “favorable” patches. We superimposed each environmental favorability map on another reflecting the known occurrence data on the respective cell to reveal the occupied and empty favorable plots, as unoccupied favorable areas must be considered in a metapopulation dynamics analysis [10]. When fragmented, these patches represented the species metapopulation structure. This process allowed to include the whole potential territory for each species, detecting those favorable cells that, having no known presence, remained hidden from the naked eye. In this way, we obtained one of the two key factors considered so far by the literature in the study of metapopulations, the patches of local populations [69]. We considered that a species distribution had a metapopulation structure when the favorable areas were patchy and the proportion of presences in areas of high favorability was greater than 0.5. In this way, generalist (very widespread) or very endemic (very territorially grouped) species were avoided.

### 2.5. Connectivity Analysis

We analyzed the connectivity between local patches or nuclei of each metapopulation [69] by focusing on the territory of intermediate and low favorability for the species (F < 0.8). This territorial matrix represents the “ocean” between the archipelago of islands to which all the patches that constitute a metapopulation [70] can be assimilated. This matrix is heterogeneous and ecologically relevant [71,72,73]. The heterogeneity resides in the variable degree of friction that different cells oppose to the passage of individuals. Thus, an index of friction should be a function reflecting how environmental variables hinder the passage through the cell, which can be computed as the complement of the environmental favorability, or 1-F, whose values are projected on each point of the territory. The cost of passage from one point to another should compute the sum of the frictions or resistance to the passage between them. This kind of analytical process is known as “cost study” [74], which assumes that an effort, work, or cost is generated in the movement from one point to another. As a general rule, the cost is proportional to the friction, and as a sum, it has a direct proportionality with distance. Therefore, the greater the distance, the greater the cost.

The calculation of the cost was performed through the r.cost function of QGIS (r. cost full for GRASS 6.0). It incorporates the value of the distance from the patches of high favorability, by summing the values of 1-F that are assigned to the grid cells that butterflies must cross to reach a cell from a favorable patch, using Knight’s neighborhood pattern (16-cells) to consider contiguous cells [74,75]. The cost value is the sum of the values of the variable assigned to each of the contiguous cells (1-F). These cost values were converted to connectivity values between 0 and 1 by computing in each cell:Connectivity = 1− [(cost − minimum cost)/(maximum cost − minimum cost)](2)

We re-scaled these values in the connectivity maps between 0 and 0.9, being 0.9 the minimum cost value obtained at the border of a favorable patch and 0 the maximum cost value, as a connectivity value of 1 was considered characteristic of the interior of the favorable patches. This procedure allowed a cartographic analysis of the territorial connectivity between the nuclei of the metapopulation of each species. For each species, a map was prepared to show the nuclei of each metapopulation structure and the connectivity of the matrix graded on a scale of eighteen levels of connectivity with a 0.05 connectivity range each to achieve a homogeneous standard for connectivity from 0 to 0.9 in all maps. Estimations and resulting maps were processed using the graphical interface of QGIS 2.18.4 [76].

## 3. Results

We obtained significant environmental favurability models for every species. Logit equations of the favorability models can be seen in Appendix A). Model discrimination was outstanding (AUC ≥ 0.9) for 139 species, excellent (0.8 ≤ AUC < 0.9) for 52 species, and acceptable (0.7 ≤ AUC < 0.8) for 31 species [52]. A total of 161 models were well-calibrated and the overall classification power of the models was acceptable, as most of the favorability models had Cohen’s kappa values higher than 0.2 and CCR values higher than 0.7.

Twenty-one species were considered not to have a metapopulation structure because their favorable areas were not fragmented. These species were *Agriades glandon*, *Araschnia levana*, *Aricia nicias*, *Boloria dia*, *B. euphrosyne*, *B. napaea*, *Cupido argiades*, *Colotis evagore*, *Erebia gorgone*, *E. oeme*, *E. pandrose*, *E. rondoui*, *Heteropterus morpheus*, *Lasiommata petropolitana*, *Minois dryas*, *Melanargia galathea*, *Pontia callidice*, *Polyommtus fulgens*, *Pyrgus cacaliae*, *Parnassius mnemosyne*, and *Satyrium ferula*.

We found that 126 species with fragmented distributions, which are listed in Appendix A, can be considered to have a metapopulation structure. The metapopulation structure and the connectivity pattern for each of them can be seen in Appendix A. The remaining 75 species distributions did not fit the metapopulation structure pattern in spite of being fragmented due to the fact that the proportion of presences in favorable areas was less than 0.5.

As examples are described here, in detail, the spatial structure of three species (Figure 2, Figure 3 and Figure 4). Figure 2A shows the cells with high environmental favorability (F ≥ 0.8) for *Argynnis adippe*, including both the cells with the presence of the species and those where the species has not been reported. The metapopulation structure of *A. adippe* has a main nucleus in the north and several nuclei scattered throughout the peninsula. Figure 2B shows the connectivity gradient among the metapopulation nuclei. The preference of *A. adippe* for high mountains (altitude > 1000 m) can be clearly seen, as high elevation metapopulation islands are connected by high connectivity bridges (in light grey) along the main mountain ranges, while low-elevation areas act as connectivity depressions (dark grey to black). The nuclei of the south are clearly isolated from those of the north and also between them by areas of low connectivity.

The metapopulation structure of *Brenthis ino* is restricted to the northern half of the Iberian Peninsula (Figure 3A). There is a main nucleus and several nuclei in the vicinity. The connectivity among them is high (Figure 3B). This is a mainland-island metapopulation structure according to [10].

*Phengaris alcon* has an interesting metapopulation structure because the highly favorable areas unoccupied for the species (grey color) are larger than the occupied high favorable areas (black colour). One very small high favorable area is in the south (Figure 4A), isolated from the northern nuclei which are more thoroughly interconnected (Figure 4B). The south favorable area is unoccupied and colonization is difficult because the connectivity with the other nuclei is very low (<0.1).

Most species present a mainland-island metapopulation structure in which the metapopulation nuclei are not isolated drastically, as the connectivity among them is generally high, >0.7 (see Appendix A). This means that if one of these smaller nuclei disappears, for any reason, it could be recolonized with individuals from the main patch. Another repeated metapopulation structure is that composed of nuclei of different sizes with no main nucleus. In this case, the possibility of colonization among the different nuclei could be more difficult because the connectivity with some nuclei is very low (<0.1) (See Appendix A).

This section may be divided by subheadings. It should provide a concise and precise description of the experimental results, their interpretation, as well as the experimental conclusions that can be drawn.

## 4. Discussion

### 4.1. The Usefulness of Fuzzy Logic for Metapopulation Studies

Fuzzy logic allowed to tackle the identification of the population patches and the computation of the connectivity among them, on a large scale such as the Iberian Peninsula and for a hyperdiverse group of species such as butterflies, which is usually hard to do. The processing of the cells’ capacity to hold populations and to permit the transit of individuals by means of a continuous function allowed to cover the whole territory, so that it could be represented graphically and processed with the analysis tools of GIS. Modeling techniques based on probability values, for example, by applying the output of the logistic regressions directly, could do this as well, but the probability is affected by the prevalence of the species, being higher for widespread species than for those with constrained distribution irrespective of the quality of the territory [14,77]. To perform a comparative analysis with a variety of species, it is necessary to use a common currency to compute the capacity of the territory to hold all the species. Fuzzy logic provided the notion of favorability, which is this common currency, in an operative way. Ref. [29] showed the usefulness of the cartographic representation of the FF, as well as the analyses derived from it, for detecting the source and sink areas for butterfly species in the Iberian Peninsula. The analysis performed shows that the capacity of the FF to assign each territorial unit with an indicative degree of the feasibility for the establishment of the different species was equally useful for identifying the metapopulation patches and the connectivity among them. For most butterflies, the east and southwest of the Iberian Peninsula were areas with very low connectivity among their metapopulation patches. These areas are abundant in sink territories (see [29]), which highlights the relationship between metapopulation theory and the source-sink theory and how fuzzy logic modeling may be useful to approach both.

The study of the dynamics of metapopulations in Ecology is classically approached from the size of the favorable patches and the distance that separates them (isolation/connectivity), which can affect the processes of extinction and recolonization between occupied and unoccupied plots through a process of emigration-immigration between them. Using the same FF, we classified for each species every cell in one of two categories: highly favorable (F ≥ 0.8), to identify the metapopulation favorable patches, and of low or intermediate favorability (F < 0.8), which constituted the inter-territorial matrix. This allowed to identify the location and size of the population patches and to establish a friction gradient through which to detect possible routes or ways of connection between the metapopulation’s patches. This kind of evaluation of the matrix characteristics complemented the concept of distance, classically considered in dispersion processes [69,78], with the concept of favorability to reflect the environmental characteristics that may facilitate or hinder migratory processes [79,80,81]. In this way, connectivity was derived from favorability and distance. This approach made the application of the metapopulation theory operational at geographic scales and for groups of species for which demographic data about birth and mortality rates and migration rates are not feasible to obtain. The mapping allowed to reflect the potential directionality of colonization movements and the possibility of the survival of propagules in a more intuitive graphical form than with a mathematical function of the distance and size of the transmitting population [82].

### 4.2. Consequences for Conservation Planning

This study has implications for the conservation of the species. The metapopulation theory puts the main conservation concern in the viability of the whole metapopulation rather than on preserving individual patches. The conservation purpose of this study is thus oriented to the maintenance of the dynamics that may allow the persistence of the metapopulation [6,10]. Butterflies in the Iberian Peninsula have distribution ranges that are heterogeneous, including areas that act as population sources or sinks, for example [29]. Determining the metapopulation patterns of species over a territory is another approach to delve into the internal complexity of the species ranges. The FF is a tool for modeling species distribution that allows considering the different roles of each part of the species distribution range [14,57,83]. The analysis of this internal complexity plays an essential role to inform the species conservation strategies. The allocation of favorability values to the inter-territorial matrix enabled a quality assessment method of the cells separating the local population centers that were beyond the presence or absence of a species’ larval host plant generally used in butterfly population studies, as [2] recommended.

Many of the cells included in the favorable patches had no reported presence of the species. In this regard, we were able to distinguish between observed (where the species has been observed to occur) and unobserved biodiversity patterns (Figure 2A, Figure 3A, Figure 4A). In this way, our analyses revealed another hidden aspect of the biogeography of Iberian butterflies, which was not possible with analysis based solely on the detected presences. As “absence of evidence is not evidence of absence” [84], for any species a blank/empty cell may represent either a true absence or an unreported presence. This distinction is important for the conservation of the species, even if both types of areas deserve to be preserved. Ref. [85] proposed that the probable distribution area of a species should include the areas of favorability higher than 0.8 (equivalent to those with odds higher than 4:1 used in their study) contiguous to areas with reported presence. Using this criterion, we may distinguish the cells with the probable unnoticed presence of the species from the likely unoccupied favorable cells. We may modify slightly the criterion of [85] for metapopulation studies by considering as probably occupied cells those being part of a set of contiguous favorable cells with at least a reported presence. This implies that patches with at least a reported presence are considered occupied in their entirety, and that unoccupied favorable patches are those contiguous favorable cells with no reported presence (Figure 2A, Figure 3A, Figure 4A).

### 4.3. Implications for a Chorological Theory

The unoccupied favorable patches are critical to understanding the dynamics of metapopulations for each species [2] and may be revealed by biogeographical modeling such as that used here. On the other hand, the colonization-extinction dynamics that occur in any species also may become visible after the modeling process, as the friction of the territory between populated patches becomes evident and the size of the patches is better estimated. All this is part of the dark biodiversity of the territory analyzed [66]. Dark biodiversity is a concept related to that of dark diversity [86,87]. While dark diversity refers to all the potential species in a territory that are not currently present, dark biodiversity refers to the degree to which the local environment interacts with the species to attract or to filter out the individuals, so driving the observed biodiversity patterns [29]. The concept of dark biodiversity allows a new approach to understanding population dynamics through the FF, resulting in the perceptibility of hidden territories where the viability of the species is possible given their environmental characteristics. Some of these territories can be vacant patches but must nevertheless be considered as belonging to the metapopulation. At the same time, the FF highlights the bridges that connect the different nuclei of the metapopulation (See Figure 2B, Figure 3B, Figure 4B, Figure 5, and Appendix A).

Patches within terrestrial ecosystems are not embedded into a homogeneous hostile matrix [2,79,80]. “The matrix matters” [88] and so, it is important to understand its characteristics [72,73]. This approach allowed us to evaluate the quality of the territorial matrix between metapopulation patches by quantifying connectivity patterns. The factors classically used for the evaluation of metapopulations so far (plot size, occupation, distance) are not questioned here, but fuzzy logic does offer a new interpretation or support since knowledge of the favorability values within the territory speaks about the chances for survival outside the places of known presence. The criterion of the distance between inhabited patches also changes after the fuzzy modeling, since the visibility of dark potential areas (areas with high favorability where the species was not reported) increases the sizes of the plots, therefore changing the proportion of occupied area and reducing the distance between them [69,89]. The size of the plots and the fraction occupied by the existing population in each are important values to establish a predictive model about the viability of the population, based mainly on its ability to emit or receive individuals [5,78,82]. Distance does not have a Euclidean influence, but an anisotropic summative effect according to the environmental characteristics between patches, which results in the connectivity [74] and reflects the likelihood of colonization of empty patches [2].

All this allows us to propose the methodology used in this paper as a predictive biogeographic tool, useful in the study of metapopulations in situations where knowledge of population demography is not possible or operational. This may facilitate the understanding of demographic processes and the dynamics generated within any metapopulation by identifying the environmental potential of the population patches and the spatial connectivity at a biogeographical scale. For example, the theory of epidemic threshold [89] considered occupied and vacant but occupiable plots, the latter being difficult to identify. This approach allowed the identification of vacant but occupiable favorable patches and the actual size of occupied patches. The effective dispersion rate (β), which is considered in the epidemic model, can be equivalent to the connectivity of the matrix while the value of the extinction rate (α) could be calculated according to the size of the occupied patches.

The resulting metapopulation patterns were very similar for some species (See Appendix A). This happened in the case of some genera (*Argynnis*, *Boloria*, *Melitaea*) or in some species with restricted geographical distribution. For example, species of the northeast quadrant or the Cantabrian Mountain (*Araschnia levana*, *Aricia nicias*, *Boloria pales*, *Cupido alcetas*, *C. argiades*, *Carcharodus palaemon*, *Erebia arvernensis*, *E. manto*, *E. pandrose*) tended to show a similar metapopulation pattern (Appendix A). These shared patterns may be a consequence of common ecological preferences of the species or of scarce evolutionary differentiation among them. For example, Figure 5 shows the connectivity map for *Argynnis aglaja*, which is very similar to that of *A. adippe* (See Figure 2B), a possible consequence of the similarity in the ecological requirements of the two species.

## 5. Conclusions

This paper showed that the FF [14] is a useful tool to reveal the metapopulation structures. Its fuzzy character allowed us to go beyond the presence-absence duality and to characterize the territory from a qualitative point of view on the basis of a series of environmental variables. This information may be relevant for the territory management and the biodiversity conservation, serving to justify the protection of new areas or the modification of contours of reserves based on the role they can play for the populations of interest. Obviously, this influences the management of resources of any kind that can be dedicated to planning or management of biodiversity.

## Figures and Tables

**Figure 1 insects-12-00392-f001:**
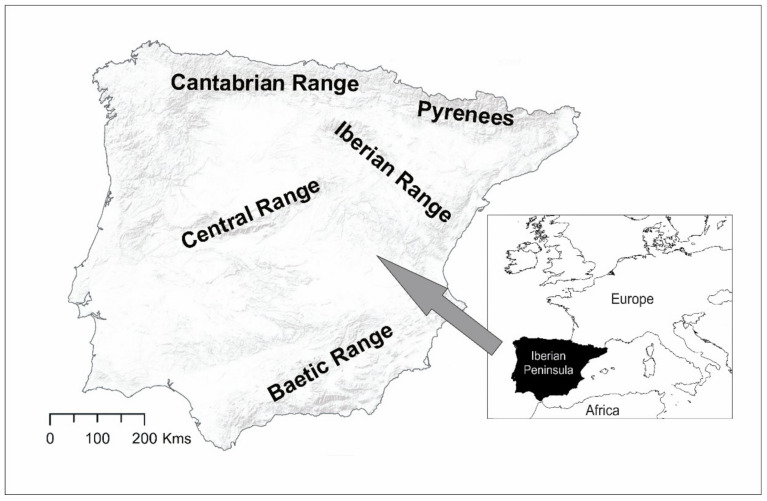
Iberian Peninsula with the main mountain ranges and the location on the European continent.

**Figure 2 insects-12-00392-f002:**
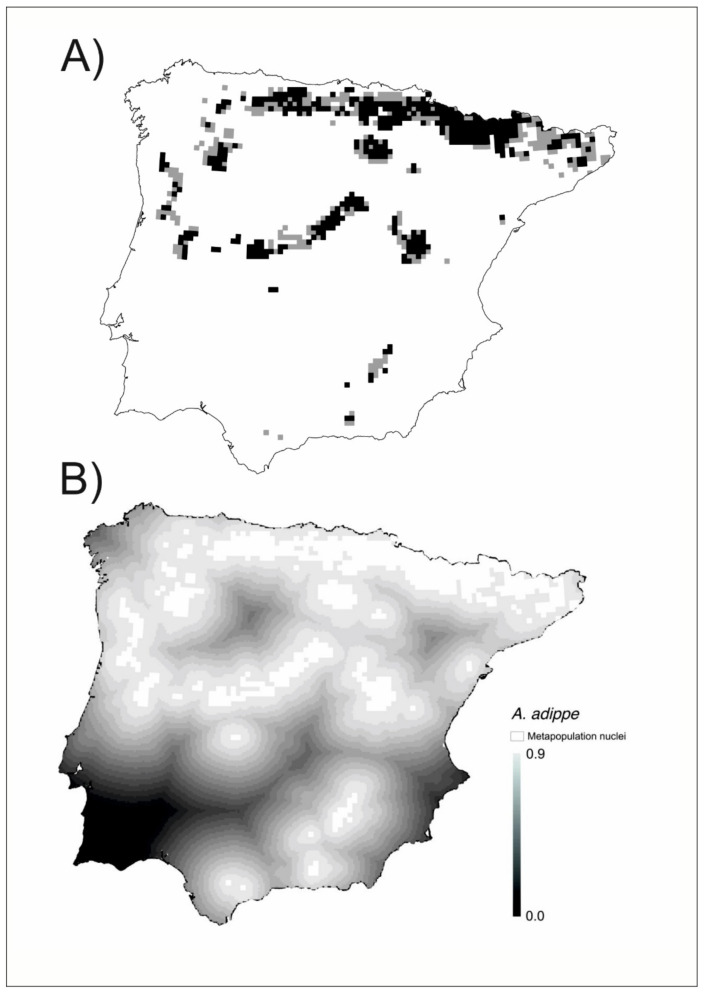
(**A**) Metapopulation patches of *Argynnis adippe*. In black, the grids with high favorability (F ≥ 0.8) and established presence, in grey those with high favorability (F ≥ 0.8) and undetected presence. (**B**) Connectivity gradient between metapopulation nuclei (white plots). The matrix has been graded on a scale of eighteen levels of connectivity with a 0.05 connectivity range each (represented in grayscale).

**Figure 3 insects-12-00392-f003:**
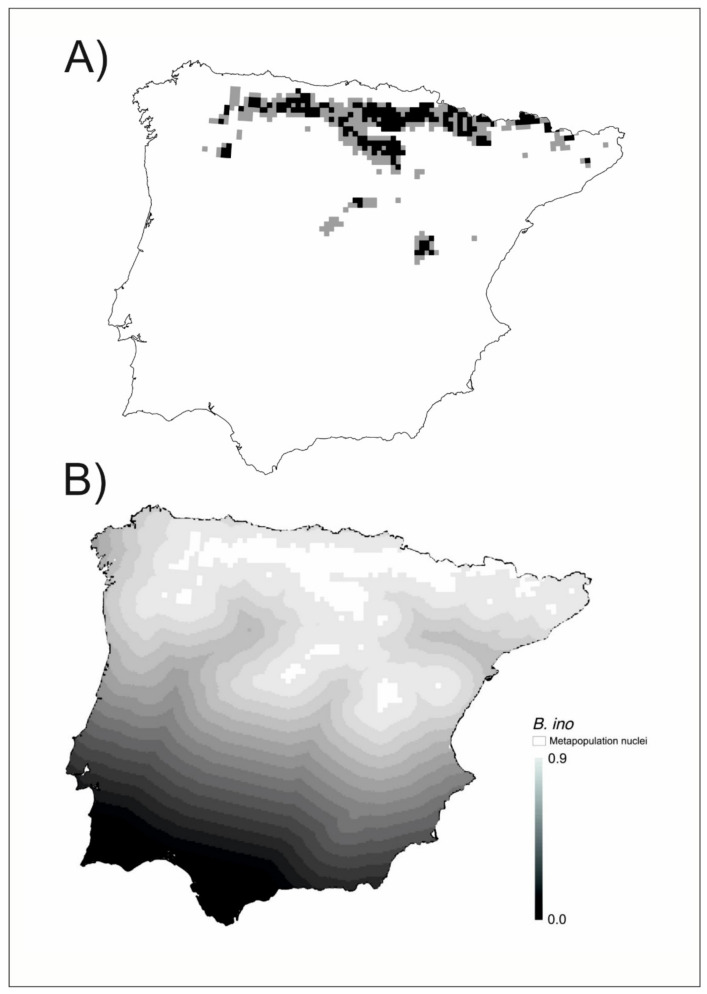
(**A**) Metapopulation patches of *Brenthis ino*. In black, the grids with high favorability (F ≥ 0.8) and established presence, in grey those with high favorability (F ≥ 0.8) and undetected presence. (**B**) Connectivity gradient between metapopulation nuclei (white plots). The matrix has been graded on a scale of eighteen levels of connectivity with a 0.05 connectivity range each (represented in grayscale).

**Figure 4 insects-12-00392-f004:**
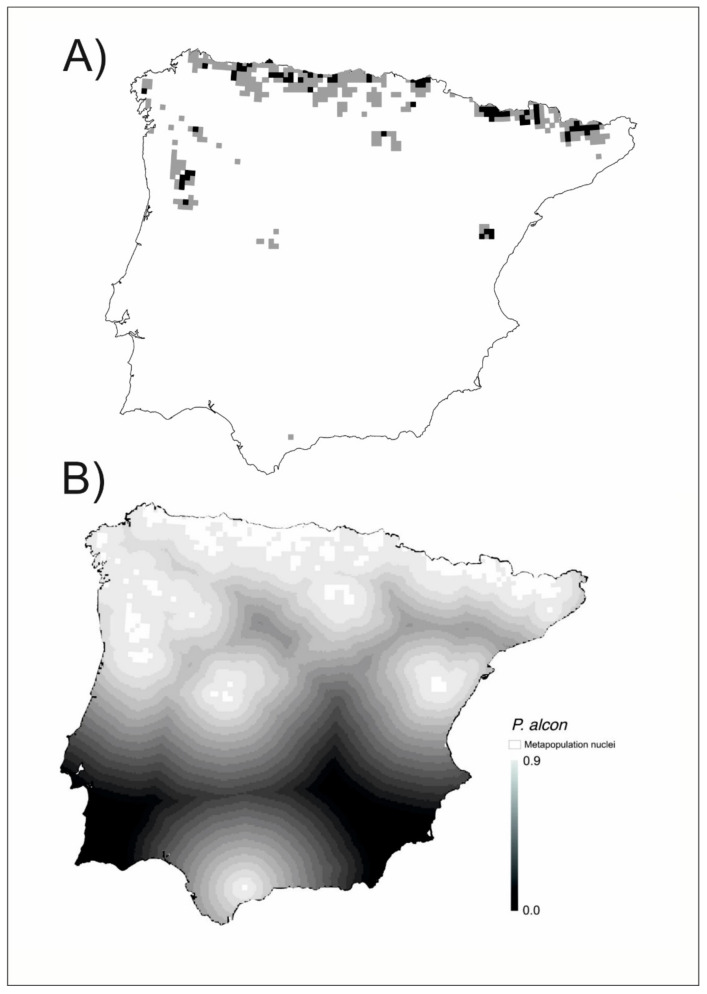
(**A**) Metapopulation patches of *Phengaris alcon*. In black, the grids with high favorability (F ≥ 0.8) and established presence, in grey those with high favorability (F ≥ 0.8) and undetected presence (**B**) Connectivity gradient between metapopulation nuclei (white plots). The matrix has been graded on a scale of eighteen levels of connectivity with a 0.05 connectivity range each (represented in grayscale).

**Figure 5 insects-12-00392-f005:**
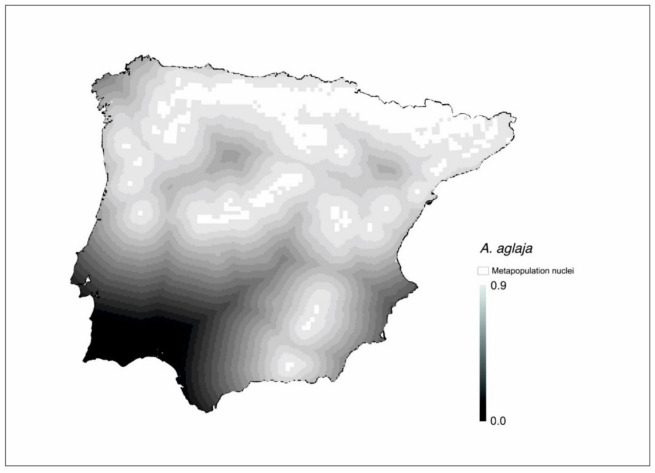
Connectivity gradient between metapopulation nuclei (white plots) of *Argynnis aglaja*. The matrix has been graded on a scale of eighteen levels of connectivity with a 0.05 connectivity range each (represented in grayscale).

## Data Availability

The data presented in this study are available on request from the corresponding author.

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
