# Peer review of "Metapopulation Patterns of Iberian Butterflies Revealed by Fuzzy Logic"

_insects, 2021, doi:10.3390/insects12050392_

Round 1

Reviewer 1 Report

1.- Lines 22-23

 Some of the favourable territories can be vacant patches but also belong to the metapopulation structure, as they may be recolonized.

2.- It may be that these territories have not been surveyed for butterflies, due to the remoteness of the entomologists' places of residence. It should not be forgotten that on very few occasions an exhaustive collection by UTM grids of territory has been carried out, and frequently the highest density of prospecting has been carried out in the residence or summer places of the Entomologists, as can be demonstrated in many of the groups studied and published.

3.- It is striking, or at least deserves an explanation, why something as elementary as the existence or not of the nutritious plant has been taken into account in this work for those Lepidopterans that are not polyphagous, it depends on the presence or not of the nutritious plant; once it has disappeared, its presence cannot exist in the area, although it could potentially be. As much as it appears in the analysis as a favorable area, it will not be until the nutrient plant reappears.

4.- And it is also striking that complete works have not been cited, of several volumes, on caterpillars and butterflies of the Iberian Peninsula in which precisely the nutritional plants and on which they make the laying occur.

5.- In appendix S3, I think the family name should be put in front of the map group and not behind it.

6.- The work seems to be of enormous interest and applicability for the study of other taxonomic groups, but I think that it would be necessary to introduce some characters that are not only climatic and geographical.

7.- some possible corrections to the text highlighted in yellow in the MS 

Line 18  “ca” perhaps it should be “can”

Line 24  “

 new areas, or the modifi-cation”  perhaps without , “new areas or the modifi-cation

Line 26 “are”, perhaps it should be “is” as referred to population (singular)

Line 27 “ by migration” “by the migration”

Line 36 “ The inverse of cost” perhaps it should be “ The inverse of the cost”

Line 38 “part of the of dark” perhaps it should be "part of the dark” without de second of

Line 38 “ butterflies, because” “ butterflies because” without ,

Line 39 “were” perhaps is “was”

Line 47 “fires or” perhaps is “fires, or

Line 51 “and certain” perhaps is and a certain

Line 64 “(local population centres)” perhaps is “(local population centers)

Line 67 “with distance between patches,” perhaps is “with the distance between patches,”

LINE 68 “that allows the recolonization” perhaps is “that allow the recolonization”

Line 74 “Fuzzy set theory, and” perhaps is “Fuzzy set theory and” without ,

Lines 80-81 “habitat transformation and of the species responses to climate change and land use modification” perhaps is better “habitat transformation, and species responses to climate change and land-use modification”

Line 90 “as case study” perhaps is better “as a case study”

Lines 109-110 “with minimum of about 450 mm per year 109 and maximum of about 2500 mm per year on average” perhaps is better “with a minimum of about 450 mm per year and a maximum of about 2500 mm per year on average”

Line 114 “peaks of much greater” perhaps is better “peaks of a much greater”

Line 124 “are of minor” perhaps is better “are of the minor”

Line 133 “spatial location and other factors.” perhaps is better “spatial location, and other factors.”

Line 135 “related to spatial situation” perhaps is better “related to the spatial situation”

Line 142 “because humans have capacity” perhaps is better “because humans have the capacity”

Line 255 “P is probability” perhaps is better “P is the probability”

Lines 256-257 “and n1 and n0 are the number of presences and absences in the dataset, respectively.” “and n1 and n0 are the number of presence and absence in the dataset, respectively.”

Line 264 “For each species we performed” perhaps is better “For each species, we performed”

Line 274 “using Akaike information” perhaps is better “using the Akaike information”

Line 283 “due to redundancy of variables” perhaps is better “due to the redundancy of variables”

Line 286 “Rate (OPR) [62] and their Cohen’s kappa” perhaps is better “Rate (OPR) [62], and their Cohen’s kappa”

Line 291 “All analyses where performed” perhaps is better “All analyses were performed”

Line 297 “In contrasts to” perhaps is better “In contrast to”

Line 311 “one the two key” perhaps is better “one of the two key”

Line 315 “In this way generalist” perhaps is better “In this way, generalist”

Line 329 “to passage between them.” perhaps is better “to the passage between them.”

Line 330 “which assumes that an effort, work or cost” perhaps is better “which assumes that an effort, work, or cost”

Line 332 “and as a sum it has a direct” perhaps is better “and as a sum, it has a direct”

Line 341 “connectivity= 1- [(cost - minimum cost)/(maximum cost - minimum cost)].” As a formula, it must be separated by paragraph' spaces anterior and posterior to de formula.

Line 347 “map was prepared showing the nuclei of each metapopulation” perhaps is better “map was prepared to show the nuclei of each metapopulation”

Line 354 “in the Appendix S1” perhaps is better “in Appendix S1”

Line 357 “total of 161 models were well calibrated” perhaps is better “total of 161 models were well-calibrated”

Line 367 “which are listed in Appendix S2” perhaps is better “which they are listed in Appendix S2”

Line 372 “As examples we describe here in detail the spatial structure” perhaps is better “As examples are described herein detail the spatial structure “

Line 374 “with presence” “with the presence”

Line 379 “along the main mountain ranges” “along with the main mountain ranges”

Line 417 “In this case the possibility” perhaps is better “In this case, the possibility”

Line 431” hyper diverse” perhaps is better “hyperdiverse”

Line 432 “The processing of the cells capacity to hold populations” perhaps is better “The processing of the cells’ capacity to hold populations”

Line 436 “but probability” “but the probability”

Line 444 “In our analysis we showed that the capacity of the FF” “The analysis performed shows that the capacity of the FF”

Line 447 “For most butterflies the east” perhaps is better “For most butterflies, the east”

Line 473 “implication” perhaps is better “implications”

Line 478 “heterogenous,” perhaps is better “heterogeneous,”

Line 488 “presences” perhaps is better “presence”

Line 491 “In this way our” “In this way, our”

Line 499 “with probable unnoticed” perhaps is better “with the probable unnoticed”

Line 507 “understand” “understanding”

Line 508 “[2], and may be” “[2] and maybe”

Line517 “understand” perhaps is better “understanding”

Line 522 “5 and Appendix S3).” perhaps is better ” 5, and Appendix S3).

Line 527 “imbedded” perhaps is better “embedded”

Line 529 “Our approach allowed to evaluate the quality of the territorial matrix” perhaps is better “This approach allowed us to evaluate the quality of the territorial matrix”

Line 534 “The criterion of distance” “The criterion of the distance”

Line 537 “size of the plots” perhaps is better “sizes of the plots”

Line 544 “proposed” perhaps is better “propose”

Lines 551-552 “Our approach allowed to identify vacant but occupiable favour-551 able patches and the actual size of occupied patches” perhaps is better “This approach allowed the identification of vacant but occupiable favourable patches and the actual size of occupied patches.”

Line 562 “be consequence” perhaps is better “be a consequence”

Line 571 “new areas, or” perhaps is better “new areas or”

Author Response

Review 1

1.- Lines 22-23

Some of the favourable territories can be vacant patches but also belong to the metapopulation structure, as they may be recolonized.

2.- It may be that these territories have not been surveyed for butterflies, due to the remoteness of the entomologists' places of residence. It should not be forgotten that on very few occasions an exhaustive collection by UTM grids of territory has been carried out, and frequently the highest density of prospecting has been carried out in the residence or summer places of the Entomologists, as can be demonstrated in many of the groups studied and published.

We agree with the reviewer. In fact, we are not saying that the absence of reported presences in a favourable patch implies that it is vacant, but that even if it is in fact vacant it has a role in the metapopulation, as it may be recolonized.

3.- It is striking, or at least deserves an explanation, why something as elementary as the existence or not of the nutritious plant has been taken into account in this work for those Lepidopterans that are not polyphagous, it depends on the presence or not of the nutritious plant; once it has disappeared, its presence cannot exist in the area, although it could potentially be. As much as it appears in the analysis as a favourable area, it will not be until the nutrient plant reappears.

We are aware of the importance of nutritious plants for the presence of monophagous butterflies. However, sometimes the knowledge of the distribution of the plant does not match that of the butterfly. This entails that the plant has not been reported in areas where the butterfly is known to be present, which makes the use of the known distribution of the plant as a predictor of butterfly presence problematic. In this work, with 222 butterfly species, it would have been too difficult to consider nutritious plants, even more given that most butterflies feed on many different plants. Monophagous species are rare. In fact, we are working on a paper about the best way to consider the nutritious plant in the distribution model of a butterfly, working with 15 monophagous species.

4.- And it is also striking that complete works have not been cited, of several volumes, on caterpillars and butterflies of the Iberian Peninsula in which precisely the nutritional plants and on which they make the laying occur.

Please see response to point 3. Since we have not used the nutritious plants in our modelling procedure we considered that it would not be necessary to add new citations to our long list of references.

5.- In appendix S3, I think the family name should be put in front of the map group and not behind it.

Done

6.- The work seems to be of enormous interest and applicability for the study of other taxonomic groups, but I think that it would be necessary to introduce some characters that are not only climatic and geographical.

We thank the reviewer for the kind comment. Please see response to point 3.

7.- some possible corrections to the text highlighted in yellow in the MS 

Line 18  “ca” perhaps it should be “can”

Done

Line 24  “  new areas, or the modifi-cation”  perhaps without , “new areas or the modifi-cation

Done

Line 26 “are”, perhaps it should be “is” as referred to population (singular)

We replaced population with populations

Line 27 “ by migration” “by the migration”

Done

Line 36 “ The inverse of cost” perhaps it should be “ The inverse of the cost”

Done

Line 38 “part of the of dark” perhaps it should be "part of the dark” without de second of

Done

Line 38 “ butterflies, because” “ butterflies because” without ,

Done

Line 39 “were” perhaps is “was”

Done

Line 47 “fires or” perhaps is “fires, or

Done

Line 51 “and certain” perhaps is and a certain

Done

Line 64 “(local population centres)” perhaps is “(local population centers)

We used the English of the UK

Line 67 “with distance between patches,” perhaps is “with the distance between patches,”

Done

LINE 68 “that allows the recolonization” perhaps is “that allow the recolonization”

Done

Line 74 “Fuzzy set theory, and” perhaps is “Fuzzy set theory and” without ,

Done

Lines 80-81 “habitat transformation and of the species responses to climate change and land use modification” perhaps is better “habitat transformation, and species responses to climate change and land-use modification”

Done

Line 90 “as case study” perhaps is better “as a case study”

Done

Lines 109-110 “with minimum of about 450 mm per year and maximum of about 2500 mm per year on average” perhaps is better “with a minimum of about 450 mm per year and a maximum of about 2500 mm per year on average”

Done

Line 114 “peaks of much greater” perhaps is better “peaks of a much greater”

Done

Line 124 “are of minor” perhaps is better “are of the minor”

We modified the sentence as follows: “Some recent minor changes in species nomenclature [33,34] do not hamper the recognition of species identities”.

Line 133 “spatial location and other factors.” perhaps is better “spatial location, and other factors.”

Done

Line 135 “related to spatial situation” perhaps is better “related to the spatial situation”

Done

Line 142 “because humans have capacity” perhaps is better “because humans have the capacity”

Don. Line 144

Line 255 “P is probability” perhaps is better “P is the probability”

Done. Line 258

Lines 256-257 “and n1 and n0 are the number of presences and absences in the dataset, respectively.” “and n1 and n0 are the number of presence and absence in the dataset, respectively.”

We reworded the sentence as follows: “n1 and n0 are the number of cells with reported presence and absence in the dataset, respectively.”. Lines 258-259

Line 264 “For each species we performed” perhaps is better “For each species, we performed”

Done. Line 267

Line 274 “using Akaike information” perhaps is better “using the Akaike information”

Done. Line 277

Line 283 “due to redundancy of variables” perhaps is better “due to the redundancy of variables”

Done. Line 286

Line 286 “Rate (OPR) [62] and their Cohen’s kappa” perhaps is better “Rate (OPR) [62], and their Cohen’s kappa” Line 290

Done

Line 291 “All analyses where performed” perhaps is better “All analyses were performed”

Done: Line 295

Line 297 “In contrasts to” perhaps is better “In contrast to”

Done. Line 301

Line 311 “one the two key” perhaps is better “one of the two key”

Done. Line 315

Line 315 “In this way generalist” perhaps is better “In this way, generalist”

Done. Line 219

Line 329 “to passage between them.” perhaps is better “to the passage between them.”

Done. Line 333

Line 330 “which assumes that an effort, work or cost” perhaps is better “which assumes that an effort, work, or cost”

Done Line 334

Line 332 “and as a sum it has a direct” perhaps is better “and as a sum, it has a direct”

Done. Line 336

Line 341 “connectivity= 1- [(cost - minimum cost)/(maximum cost - minimum cost)].” As a formula, it must be separated by paragraph' spaces anterior and posterior to de formula.

Done Line 345-346

Line 347 “map was prepared showing the nuclei of each metapopulation” perhaps is better “map was prepared to show the nuclei of each metapopulation”

Done. Line 353

Line 354 “in the Appendix S1” perhaps is better “in Appendix S1”

Done. Line 360. We added Table 1 to this appendix S1, so the logits equations are now Table 2.

Line 357 “total of 161 models were well calibrated” perhaps is better “total of 161 models were well-calibrated”

Done. Line 363

Line 367 “which are listed in Appendix S2” perhaps is better “which they are listed in Appendix S2”

We rephrased the sentence as follows: “We found that 126 species with fragmented distributions, which are listed in Appendix S2 (Supporting Information), can be considered to have a metapopulation structure.” Line 372-374

Line 372 “As examples we describe here in detail the spatial structure” perhaps is better “As examples are described herein detail the spatial structure “

We modified the sentence as: “As examples are described here, in detail, the spatial structure…” Line 379-380

Line 374 “with presence” “with the presence”

Done. Line 382

Line 379 “along the main mountain ranges” “along with the main mountain ranges”

We modified the text as follows: “The preference of A. adippe for high mountains (altitude > 1000 m) can be clearly seen, as high elevation metapopulation islands are connected by high connectivity bridges (in light grey) along the main mountain ranges, while low-elevation areas act as connectivity depressions (dark grey to black).”. Line 385-389

Line 417 “In this case the possibility” perhaps is better “In this case, the possibility”

Done. Line 425

Line 431” hyper diverse” perhaps is better “hyperdiverse”

Done. Line 439

Line 432 “The processing of the cells capacity to hold populations” perhaps is better “The processing of the cells’ capacity to hold populations” 

Done. Line 440

Line 436 “but probability” “but the probability”

Done. Line 444

Line 444 “In our analysis we showed that the capacity of the FF” “The analysis performed shows that the capacity of the FF”

Done. Line 452

Line 447 “For most butterflies the east” perhaps is better “For most butterflies, the east”

Done. Line 456

Line 473 “implication” perhaps is better “implications”

Done. Line 481

Line 478 “heterogenous,” perhaps is better “heterogeneous,”

Done. Line 486

Line 488 “presences” perhaps is better “presence”

Done. Line 496

Line 491 “In this way our” “In this way, our”

Done. Line 499

Line 499 “with probable unnoticed” perhaps is better “with the probable unnoticed”

Done. Line 507

Line 507 “understand” “understanding”

Done. Line 515

Line 508 “[2], and may be” “[2] and maybe”

Done. Line 516

Line517 “understand” perhaps is better “understanding”

Done. Line 525

Line 522 “5 and Appendix S3).” perhaps is better ” 5, and Appendix S3).

Done. Line 530

Line 527 “imbedded” perhaps is better “embedded”

Done Line 535

Line 529 “Our approach allowed to evaluate the quality of the territorial matrix” perhaps is better “This approach allowed us to evaluate the quality of the territorial matrix”

Done. Line 537

Line 534 “The criterion of distance” “The criterion of the distance”

Done. Line 543

Line 537 “size of the plots” perhaps is better “sizes of the plots”

Done. Line 545

Line 544 “proposed” perhaps is better “propose”

Done. Line 553

Lines 551-552 “Our approach allowed to identify vacant but occupiable favour- able patches and the actual size of occupied patches” perhaps is better “This approach allowed the identification of vacant but occupiable favourable patches and the actual size of occupied patches.”

Done. Line 560

Line 562 “be consequence” perhaps is better “be a consequence”

Done. Line 571

Line 571 “new areas, or” perhaps is better “new areas or”

Done. Line 580

Reviewer 2 Report

This manuscript analyzes the distribution of butterflies throughout the whole Iberian peninsula,  attempting to reveal their potential metapopolational structure.The Authors overcome the limitations of presence/absence data by using a favourability index of occurrence, matching habitat features of a 10x10Km grid, to a  favourability function. Known occurrence data reveal the occupied and empty favourable patches, as unoccupied patches are functional in a metapopulation dynamics analysis. The aim is to obtain a map that describes the territorial connections identified through a favorability criterion potentially useful to highlight a metapopulational structure, intended as a component of the dark biodiversity. Detection of the favorable patches for Iberian butterfly species is expected to be a guide to environmental planning for conservation. By this approach the Authors reveal a potential metapopulational structure in 126 out of 220 butterfly species. The work is clearly written in aims, methodology and results and utilizes a novel approach based on the fuzzy set theory that could also be tested on other organisms.

In my view It is a valuable approach and I strongly recommend its publication in Insects. 

A few minor adjustments would be advisable. Table 1 is very long and should be better placed in supplementary information. The variables employed were apparently the same used and published in a previous paper (i.e.: Pulido-Pastor et al., Insect Conserv. Divers. 2018), where also the basic niche modelling was performed. I would expect that such partly cover will be more clearly accounted in the text.

The discussion in section 4.3 is overall clear, yet the title referring to a “chorological theory” looks not well fitting. Unless the Authors explain better their view, I suggest changing it since chorological theory according to the most shared sense of view concerns distribution models, usually at the scale of the species, and is an aspect of "Systematic Biogeography". E.g.: see Fattorini, S. (2016). A history of chorological categories. History and Philosophy of the Life Sciences, 38 (3), 1-21.

Author Response

Review 2

A few minor adjustments would be advisable. Table 1 is very long and should be better placed in supplementary information.

Done. Line 137 and Lines 584-585

The variables employed were apparently the same used and published in a previous paper (i.e.: Pulido-Pastor et al., Insect Conserv. Divers. 2018), where also the basic niche modelling was performed. I would expect that such partly cover will be more clearly accounted in the text.

We added the following sentence at the beginning of the paragraph devoted to predictor variables: “We used the same variables previously used to assess the source-sink dynamics of butterflies in the Iberian Peninsula [29].” Lines 126-127

We modified the sentence at the beginning of the section 2.3 as follows: “We used the same favourability models used to analyse the source-sink dynamics of butterfly species in the Iberian Peninsula [29].” Line 251

The discussion in section 4.3 is overall clear, yet the title referring to a “chorological theory” looks not well fitting. Unless the Authors explain better their view, I suggest changing it since chorological theory according to the most shared sense of view concerns distribution models, usually at the scale of the species, and is an aspect of "Systematic Biogeography". E.g.: see Fattorini, S. (2016). A history of chorological categories. History and Philosophy of the Life Sciences, 38 (3), 1-21.

We agree with the reviewer that chorological theory concerns distribution models, usually at the scale of the species, and is an aspect of "Systematic Biogeography", and we feel that our work fits this definition. This is why we preferred to maintain the subtitle.
